# SRON: State-free LLM Training via Row-wise Gradient Normalization

## Abstract

Large Language Models (LLMs) achieve remarkable performance but at the cost of substantial memory overhead, particularly when trained with memory-intensive adaptive optimizers such as Adam. As model sizes continue to grow, memory efficiency has become a critical bottleneck. Existing approaches often rely on costly techniques such as singular value decomposition or matrix-level operations to reduce memory usage, which can slow training or degrade performance. In this paper, we propose **SGD with Row-wise Normalization (SRON)**, a state-free optimizer motivated by observed row-level gradient disparities in the Attention module. We provide a theoretical analysis establishing SRON's convergence under non-convex $L$-Lipschitz smoothness conditions, ensuring its soundness for large-scale models. Extensive experiments across architectures (LLaMA, GPT, Gemma) and model sizes (60M–7B parameters) show that SRON reduces optimizer state memory overhead by 90%–100% and cuts training time by up to 67% on billion-parameter models. Moreover, SRON consistently matches or outperforms Adam and other baselines on both pre-training and fine-tuning tasks, demonstrating its effectiveness as a memory-efficient and high-performance optimizer for LLM training.

## 1 Introduction

Large Language Models (LLMs) have made significant strides across a variety of domains (Brown et al., 2020; Touvron et al., 2023b). Their remarkable success can be attributed to their massive parameter sizes and the vast amounts of training data they are exposed to, which together enable them to demonstrate exceptional inductive reasoning capabilities, often surpassing traditional models.

Despite their success, training such complex models presents substantial challenges. While Stochastic Gradient Descent (SGD) (Bottou, 2010) is both memory-efficient and conducive to rapid training, its lack of adaptive mechanisms limits its effectiveness when applied to sophisticated architectures like Transformers (Vaswani et al., 2017), particularly LLMs (Zhang et al., 2020; Kunstner et al., 2023; 2024). In this context, adaptive optimization algorithms have become increasingly essential for efficient LLM training. Among these, Adam(W) (Kingma & Ba, 2014; Loshchilov & Hutter, 2017) has emerged as the dominant choice, owing to its powerful combination of first-order momentum for acceleration and second-order moment estimates that facilitate per-parameter learning rate adjustments.

However, as model sizes grow, the memory demands of Adam become a critical bottleneck, limiting scalability. Adam maintains both first- and second-order moment estimates for each parameter, effectively doubling memory requirements compared to the model parameters alone. For example, training the LLaMA-7B (Touvron et al., 2023b) model with Adam consumes over 28 GB of optimizer state memory in BF16 precision (Zhao et al., 2024), while for GPT-3 (Brown et al., 2020), this can exceed 700 GB. Such memory overhead underscores the need for optimizers that balance memory efficiency and model performance.

Recent research has focused on reducing optimizer memory usage, primarily through two strategies: (1) minimizing redundancy in the optimizer state and (2) preprocessing gradients to improve efficiency.

**Reducing State Redundancy in Adam.** Methods in this category aim to identify and eliminate redundant components in Adam's optimizer state. For instance, GaLore (Zhao et al., 2024) applies Singular Value Decomposition (SVD) to project gradients into a lower-dimensional subspace, while Fira (Chen et al., 2024) introduces residuals to mitigate information loss from projection. These approaches, however, rely on computationally expensive SVD operations that scale poorly with model size. APOLLO addresses this by replacing SVD with random projections, and other methods like Adam-Mini (Zhang et al., 2024) and Adam-S (Zhang et al., 2025) optimize memory through block-wise sharing or momentum-based adaptive rates, respectively.

**Gradient Preprocessing.** This strategy modifies gradients to reduce memory and training stability. MUON (Liu et al., 2025) uses Newton-Schulz (N-S) iterations to orthogonalize first-order momentum, enabling adaptive learning rates without storing second-order moments. SWAN (Ma et al., 2024) introduces gradient norm normalization and fast N-S whitening to enable stateless training. While effective, these methods depend on costly matrix-level operations. SinkGD (Scetbon et al., 2025) mitigates this issue by employing multi-norm normalization with gradient projections, thereby reducing computational overhead. However, since these methods require preprocessing the entire gradient matrix, they suffer from significant communication costs.

These limitations highlight the need for a simpler, memory-efficient approach that adaptively scales gradients without relying on expensive matrix computations. Motivated by this, we propose **SGD with Row-wise Normalization (SRON)**, a variant of SGD (Bottou, 2010) that adjusts learning rates on a row-wise basis for two-dimensional gradients. Unlike existing methods that normalize the entire gradient as a 1D tensor, SRON is structure-aware, preserving the inherent gradient organization. Our **key insight** is that row-wise gradient scales in LLM training vary significantly, largely due to sparsity induced by attention mechanisms. SRON applies structured row-wise normalization to rescale gradients, ensuring comparable magnitudes across all rows. This enables effective training with plain SGD—without auxiliary optimizer states—substantially reducing memory overhead. Empirically, SRON achieves performance on par with or exceeding Adam while offering superior memory efficiency, often matching or surpassing other memory-efficient optimizers.

Our contributions are summarized as follows:

- **Memory-Efficient, State-free Optimizer:** We propose SRON, a well-motivated, simple, effective, and state-free optimizer. Compared with Adam, SRON reduces total training memory by 64% and optimizer state memory by 90%, cuts training time by 67%, and improves model performance.
- **Theoretical Insights and Guarantees:** Within a simplified Transformer framework, we analyze row-wise scale correlations in two-dimensional gradients, identifying sources of extreme variance. We provide convergence guarantees for SRON under both standard non-convex $L$-Lipschitz smoothness, grounding its effectiveness in modern LLM architectures.
- **Extensive Empirical Validation:** Across models from 60M to 7B parameters, including LLaMA, GPT-2, and Gemma, SRON consistently improves model quality, reduces memory usage, and accelerates training in both pre-training and fine-tuning settings, demonstrating strong practicality and robustness against competitive baselines.

## 2 RELATED WORKS

### 2.1 GRADIENT NORMALIZATION

Gradient normalization has emerged as an alternative to gradient clipping (Pascanu et al., 2013) for stabilizing optimization. Unlike simple clipping, which directly limits gradient values, normalization methods adjust gradients at various stages of the optimization process. For example, Batch Norm (Ioffe & Szegedy, 2015) and Layer Norm (Ba et al., 2016) normalize the activation values, thereby influencing the gradient distribution to a certain degree, while adaptive large-batch optimizers such as LARS (You et al., 2017) and LAMB (You et al., 2019) scale parameter updates based on the gradient norms.

Some methods reinterpret normalization in novel ways. Adam (Kingma & Ba, 2014) normalizes the variance by element-wise storage of the second-order moments of gradients. SignSGD (Bernstein

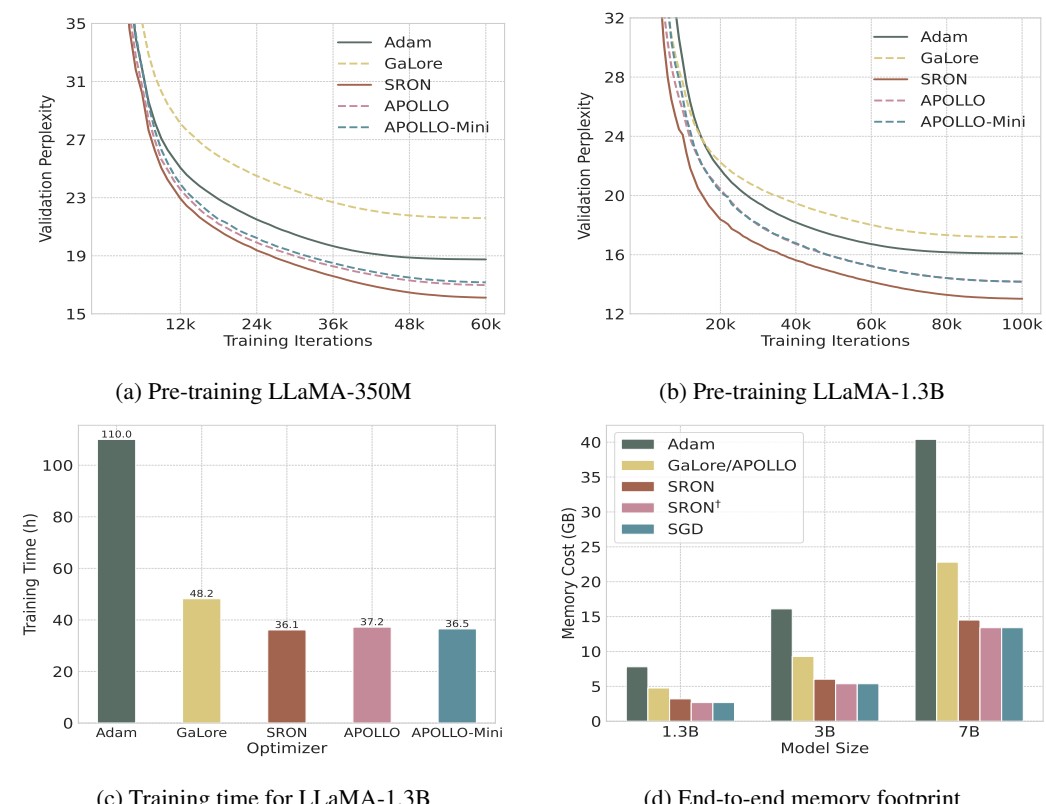

(a) Pre-training LLaMA-350M

(b) Pre-training LLaMA-1.3B

(c) Training time for LLaMA-1.3B

(d) End-to-end memory footprint

Figure 1: **SRON performance preview on LLM pre-training.** Figures (a) and (b) show pre-training results on LLaMA-350M and LLaMA-1.3B on the C4 dataset, where SRON consistently achieves lower validation perplexity than competing baselines. Figure (c) presents the wall-clock training time for the LLaMA-1.3B model on 32 NVIDIA RTX 3090 24GB GPUs. Compared with Adam, SRON cuts total training time by 67%. Figure (d) reports the memory footprint (model parameters + optimizer states, in BF16) during LLM pre-training. Compared to Adam, SRON reduces the total training memory by approximately 64% when pre-training a 7B model. Moreover, SRON† eliminates optimizer state storage entirely, resulting in memory overhead equivalent to that of SGD without momentum.

et al., 2018) discards the gradient magnitudes entirely, focusing solely on their signs. In large-scale training scenarios, such as for LLMs, normalization-based strategies are particularly effective. For instance, Lion (Chen et al., 2023) improves upon SGD by using a more refined sign function. SWAN (Ma et al., 2024) applies whitening to normalized gradients, while SinkGD (Scetbon et al., 2025) extends this normalization approach across multiple norms. Collectively, these methods highlight the importance of normalized SGD variants for stabilizing and scaling optimization in modern deep learning models.

## 2.2 MEMORY-EFFICIENT OPTIMIZATION

Memory-efficient optimization strategies can be broadly categorized into two approaches. The first focuses on reducing the number of trainable parameters by freezing them, while the second focuses on designing optimizers with minimal or no auxiliary optimizer states.

The LoRA method (Hu et al., 2022) freezes pre-trained parameters and uses low-rank approximation to update the remaining parameters. ReLoRA (Lialin et al., 2023) applies LoRA updates indirectly to the frozen parameters. FLoRA (Hao et al., 2024) introduces random projections to further improve memory efficiency. Numerous variants and extensions of LoRA (Zhang et al., 2023; Xia et al., 2024; Dettmers et al., 2024) have also been proposed to enhance its performance. BAdam (Luo et al., 2024) employees block coordinate descent to Adam's update rule. In contrast, GaLore (Zhao

et al., 2024) compresses gradients via SVD before updating the optimizer states. This approach has inspired several follow-up works: APOLLO (Zhu et al., 2024) replaces SVD with random projections for improved efficiency, GWT (Wen et al., 2025) applies wavelet transforms, Adam-Mini (Zhang et al., 2024) reduces memory through block-wise approximations of second-order moments, and RSO (Chen et al., 2025) decomposes the original training problem for optimizer and activation memory. More recently, MUON (Liu et al., 2025) achieves adaptive learning rates by leveraging only momentum and matrix orthogonalization. Additionally, methods based on gradient normalization, such as SWAN (Ma et al., 2024) and SinkGD (Scetbon et al., 2025), enable SGD-style updates that further reduce memory overhead.

In summary, the two research directions—gradient normalization and memory-efficient optimization—offer complementary approaches for enhancing stability and scalability in large-scale models. By combining insights from both, our SRON method provides a memory-efficient, stable training mechanism for large models without relying on auxiliary optimizer states.

Table 1: **Memory and complexity comparison of different methods.** Suppose $\mathbf{W} \in \mathbb{R}^{m \times n} (m \leq n)$, GaLore, APOLLO adopt a rank of $r$.

|  | Adam | MUON | Adam-Mini | GaLore | APOLLO | SWAN | **SRON** |
|---|---|---|---|---|---|---|---|
| Memory | $2mn$ | $mn$ | $mn$ | $mr + 2nr$ | $mr + 2nr$ | $0$ | $0$ |
| Complexity | $\mathcal{O}(mn)$ | $\mathcal{O}(m^2n)$ | $\mathcal{O}(mn)$ | $\mathcal{O}(m^2n)$ | $\mathcal{O}(mnr)$ | $\mathcal{O}(m^2(m+n))$ | $\mathcal{O}(mn)$ |

## 3 PRELIMINARIES

### 3.1 NOTATION

We denote scalars/vectors by lower-case/lower-case boldface letters. We denote matrices by upper-case boldface letters. For $\mathbf{G} = (\mathbf{g}_1, \mathbf{g}_2, \dots, \mathbf{g}_m)^T \in \mathbb{R}^{m \times n}$, we use $\|\mathbf{G}\|$ to denote its Frobenius-norm, $[\mathbf{G}]_i$ to denote its $i$-th column, diag$\{\dots\}$ to represnt a diagonal matrix, and $\odot$ to repesent the element-wise multiplication. $f : \mathbb{R}^{m \times n} \to \mathbb{R}$ represents the loss function, and $\mathbf{W}$ to denote the model parameters.

### 3.2 VANILLA ADAM OPTIMIZER

Adam(W) (Kingma & Ba, 2014; Loshchilov & Hutter, 2017) has become the standard optimizer for LLMs. Adam updates the model parameters $\mathbf{W} \in \mathbb{R}^{m \times n}$ as follows

$$\mathbf{W}^{t+1} = \mathbf{W}^t - \eta \tilde{\mathbf{G}}^t, \quad \tilde{\mathbf{G}}^t = \mathbf{M}^t \odot \frac{1}{\sqrt{\mathbf{V}^t} + \epsilon}, \tag{1}$$

where $\eta$ is the learning rate ($lr$), and $\epsilon$ is a small constant for numerical stability. The first moment $\mathbf{M}^t$ and the second moment $\mathbf{V}^t$ are computed as exponentially moving averages

$$\mathbf{M}^{t+1} = \beta_1 \mathbf{M}^t + (1 - \beta_1)\mathbf{G}^t, \quad \mathbf{V}^{t+1} = \beta_2 \mathbf{V}^t + (1 - \beta_2)(\mathbf{G}^t)^{\odot 2},$$

where $\mathbf{G}^t$ denotes the batch gradient at time step $t$, and $\beta_1, \beta_2 \in [0, 1)$ are the exponential decay rates. Adam leverages the first moment $\mathbf{M}^t$ to smooth the update direction, eliminating the noise from mini-batches (Zhang et al., 2020; Cutkosky & Mehta, 2020), while using the second moment $\mathbf{V}^t$ to customize adaptive learning rates for each element. The second-order moments approximate the diagonal elements of the Fisher information matrix (Kingma & Ba, 2014; Hwang, 2024), effectively providing an approximate whitening of the gradients (Ma et al., 2024).

Despite its advantages, Adam comes with significant memory overhead due to the need to track both first- and second-order moment estimates. This element-wise tracking requires storing twice the model's size in the optimizer state ($\mathbf{M}, \mathbf{V} \in \mathbb{R}^{m \times n}$), which has become a substantial bottleneck, especially when training large-scale LLMs.

## 4 MOTIVATION AND ALGORITHM

In this section, we present the design motivation behind our proposed optimizer, which applies row-wise normalization to gradients prior to parameter updates.

Our motivation stems from discrepancies in gradient row-norm scales observed in the attention modules during LLM training. To investigate this, we pre-train LLaMA-60M and LLaMA-130M on the C4 dataset and record the maximum ratio of row norms across gradients in the shallowest and deepest attention layers, defined as $\max_{1 \leq i \leq m} \|\mathbf{g}_i\| / \min_i \|\mathbf{g}_i\|$. We focus on the gradients of the query, key, value, and output projections (*q_proj*, *k_proj*, *v_proj*, and *o_proj*), with results shown in Figure 2 and Figure 5 (Appendix).

In the early stages of training, the row-norm ratios of the query, key, and value projections exhibit sharp fluctuations, with extreme cases exceeding a 500-fold difference. As training progresses, these fluctuations gradually stabilize, indicating increasingly steady gradient dynamics. However, the larger LLaMA-130M model displays more severe gradient oscillations than the 60M model in the early phase, particularly in shallow-layer attention modules. This extreme variability highlights the stringent demands placed on optimization strategies and motivates the need for more robust, structure-aware approaches such as SRON.

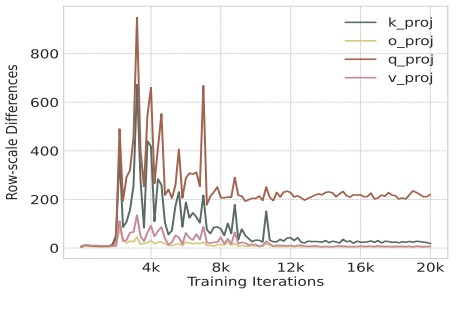 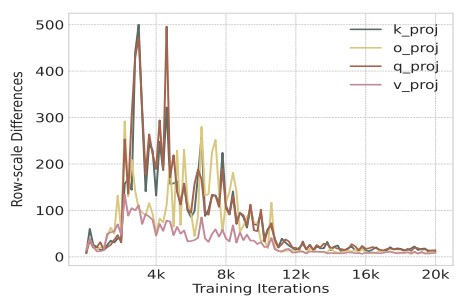

(a) Layer-0, LLaMA-130M  (b) Layer-11, LLaMA-130M

Figure 2: Row-norm scale differences of gradients in attention modules (query, key, value, and output projections) on LLaMA-130M.

A common solution to this issue is to normalize the gradients, scaling the updates to a common magnitude. Existing normalization methods often flatten high-dimensional gradients into a one-dimensional vector for uniform normalization, which ignores the intrinsic structural information of the gradients. When the value of a particular dimension is too large, it suppresses the updates of other dimensions. Based on this observation, we propose a more targeted approach: row-wise normalization. Consider $\mathbf{G}^t = (\mathbf{g}_1^t, \mathbf{g}_2^t, \ldots, \mathbf{g}_m^t)^T \in \mathbb{R}^{m \times n}$, with $\{\mathbf{g}_i^t\}_{1 \leq i \leq m}$ representing the rows, we normalize the gradient by its row-wise second-order moment

$$\mathbf{V}_{i,i}^t = \left( \sqrt{\frac{1}{n} \sum_{j=1}^{n} (\mathbf{G}_{i,j}^t)^2} + \epsilon \right)^{-1}, \quad \tilde{\mathbf{G}}^t = \mathbf{V}^t \mathbf{G}^t, \tag{2}$$

where $\mathbf{V}^t \in \mathbb{R}^{m \times m}$, and $\epsilon > 0$ for numerical stability. Therefore, we update the weights by the standard SGD recursion

$$\mathbf{W}^{t+1} = \mathbf{W}^t - \eta_t \tilde{\mathbf{G}}^t. \tag{3}$$

This results in **SGD with Row-wise Normalization (SRON)** in Algorithm 1.

## 5 CONVERGENCE GUARANTEES

We first establish convergence results for general non-convex functions under standard smoothness and stochastic gradient assumptions.

**Assumption 5.1** (*L*-smoothness)**.** The loss function $f$ has an $L$-Lipschitz continuous gradient, i.e.,

$$\|\nabla f(\mathbf{W}) - \nabla f(\mathbf{W}')\| \leq L \|\mathbf{W} - \mathbf{W}'\|, \quad \forall \, \mathbf{W}, \mathbf{W}' \in \mathbb{R}^{m \times n}.$$

---

**Algorithm 1** SRON Optimizer

---

**Input:** Weight matrix $\mathbf{W}$, step size $\eta$, batch size $b$, number of iterations $T$, numerical stability parameter $\epsilon$, scaling coefficient $\alpha$.

Initialize $t \leftarrow 0$

**repeat**

$\quad \mathbf{G}_t \leftarrow \frac{1}{b} \sum_{i=1}^{b} \nabla_{\mathbf{W}^i} f_i(\mathbf{W}^i)$ $\hfill$ {Batch gradient}

$\quad \mathbf{V}^t \leftarrow \text{diag} \left\{ \left( \frac{1}{n} \sum_{j=1}^{n} (\mathbf{g}_{i,j}^t)^2 + \epsilon \right)^{-1} \right\}_{1 \leq i \leq m}$

$\hfill$ {Compute row-wise normalizer}

$\quad \tilde{\mathbf{G}}^t \leftarrow \mathbf{V}^t \mathbf{G}^t$ $\hfill$ {Normalize gradient row-wise}

$\quad \mathbf{W}^{t+1} \leftarrow \mathbf{W}^t - \alpha \eta_t \tilde{\mathbf{G}}^t$ $\hfill$ {Update weights}

$\quad t \leftarrow t + 1$

**until** $t = T$

---

**Assumption 5.2** (Bounded gradient). There exists a constant $M \geq 0$ such that

$$\|\mathbf{G}^t\| \leq M, \quad \forall\, t.$$

**Assumption 5.3** (Unbiased estimate with bounded variance). The stochastic gradient $\mathbf{G}^t$ satisfies

$$\mathbb{E}\left[\mathbf{G}^t \mid \mathcal{F}^t\right] = \nabla f(\mathbf{W}^t),$$
$$\mathbb{E}\left[\|\mathbf{G}^t - \nabla f(\mathbf{W}^t)\|^2 \mid \mathcal{F}^t\right] \leq \sigma^2, \quad \forall\, t,$$

where $\mathcal{F}^t$ denotes the history up to step $t$.

These assumptions are standard in stochastic optimization and provide the foundation for establishing convergence guarantees.

**Theorem 5.4** (Convergence Under $L$-smoothness). *Let $\{\mathbf{W}^t\}_{t \geq 1}$ be generated by Algorithm 1. Suppose Assumptions 5.1–5.3 hold, and let the step size be $\eta_t = \eta_0/\sqrt{t}$. Then,*

$$\min_{1 \leq t \leq T} \mathbb{E}\left[\|\nabla f(\mathbf{W}^t)\|^2\right] = \mathcal{O}\left(\frac{\ln T}{\sqrt{T}}\right).$$

*The hidden constant in $\mathcal{O}(\cdot)$ depends on $L, M, \sigma, m, n, \epsilon$, and $\eta_0$.*

The rate $\mathcal{O}\left(\frac{\ln T}{\sqrt{T}}\right)$ matches that of other adaptive stochastic optimization methods (Reddi et al., 2019; Zhou et al., 2024), thus providing a theoretical guarantee for SRON under standard smoothness conditions.

## 6 NUMERICAL RESULTS

In this section, we empirically validate the effectiveness of the proposed SRON optimizer, with a primary focus on large language model (LLM) training tasks. Specifically, we pre-train LLaMA models (Touvron et al., 2023b) of varying sizes on the English portion of the Colossal Clean Crawled Corpus (C4) dataset (Raffel et al., 2019), access SRON in models beyond LLaMA, and conduct ablation studies on the normalization methods. Detailed experimental configurations, hyperparameters, and computational environments are provided in Appendix C.

### 6.1 MEMORY-EFFICIENT PRE-TRAINING

**Setup.** We pre-train LLaMA models ranging from 60M to 7B parameters on C4, following the training configuration of Zhao et al. (2024). The total batch size is 512 with a sequence length of 256. A linear warm-up is applied for the first 10% of training steps, followed by a cosine decay to 10% of the peak learning rate. All experiments use BF16 precision to reduce memory consumption and are parallelized using Distributed Data Parallel (DDP) across multiple GPUs with gradient synchronization using PyTorch's (Paszke et al., 2017) *torch.distributed* framework.

Table 2: **Comparison of memory-efficient training methods for pre-training LLaMA models (60M–1.3B) on the C4 dataset.** We report the final validation PPL and the estimated memory usage of optimizer states. Results marked with $^*$ indicate values reported in prior works.

| Methods | 60M | | 130M | | 350M | | 1.3B | |
|---|---|---|---|---|---|---|---|---|
| | Perplexity | Memory | Perplexity | Memory | Perplexity | Memory | Perplexity | Memory |
| Adam | 30.05 | 0.22G | 24.95 | 0.53G | 18.75 | 1.47G | 16.10 | 5.12G |
| MUON | 28.93 | 0.18G | 23.05 | 0.36G | 16.96 | 0.86G | 14.28 | 2.93G |
| GaLore | 34.38 | 0.16G | 26.47 | 0.30G | 19.36 | 0.64G | 15.66 | 2.08G |
| APOLLO | 31.26 | 0.16G | 23.35 | 0.30G | 16.73 | 0.64G | 14.20 | 2.08G |
| APOLLO-Mini | 31.58 | 0.12G | 23.83 | 0.19G | 17.17 | 0.26G | 14.18 | 0.52G |
| SWAN$^*$ | 30.59 | 0.12G | 22.61 | 0.19G | 16.63 | 0.26G | 13.56 | 0.52G |
| SinkGD$^*$ | 30.99 | 0.12G | 22.75 | 0.19G | 16.51 | 0.26G | 13.51 | 0.52G |
| **SRON** | **29.91** | 0.12G | **22.51** | 0.19G | **16.11** | 0.26G | **13.02** | 0.52G |
| SGD | 736.9 | 0.00G | 640.9 | 0.00G | 398.7 | 0.00G | 276.3 | 0.00G |
| SignSGD | 51.12 | 0.00G | 40.33 | 0.00G | 31.36 | 0.00G | 26.72 | 0.00G |
| **SRON$^\dagger$** | **39.25** | 0.00G | **28.48** | 0.00G | **19.83** | 0.00G | **16.79** | 0.00G |
| Training Tokens | 1.3B | | 2.6B | | 7.8B | | 13.1B | |

**Baselines.** For comparative analysis, we evaluate the following baseline optimizers: **Adam** (Kingma & Ba, 2014), the standard optimizer widely used for training large models. **MUON** (Liu et al., 2025), which orthogonalizes gradient momentum with N-S iteration. **GaLore** (Zhao et al., 2024), a memory-efficient variant of Adam that leverages low-rank gradient projections. **APOLLO** (Zhu et al., 2024), Adam with random projections. **APOLLO-Mini** (Zhu et al., 2024), a variant of APOLLO with rank $r = 1$ for further memory reduction. **SGD** (Bottou, 2010), the standard gradient descent method with minimal memory usage and no gradient processing. **SignSGD** (Bernstein et al., 2018), which applies the sign function to gradients before updating parameters. **SWAN** (Ma et al., 2024) and **SinkGD** (Scetbon et al., 2025), due to the lack of publicly available implementations and the identical experimental setup, we report the results as presented in the original publication.

To enable a fair comparison with memory-efficient baselines, we define two variants of SRON:

**SRON**: Motivated by the design principles of SRON, we apply SRON updates exclusively to parameters in linear projection modules, such as attention and MLP layers, while optimizing all other parameters using Adam. This hybrid optimization scheme is commonly adopted in memory-efficient optimizers (Zhao et al., 2024; Zhu et al., 2024; Ma et al., 2024; Liu et al., 2025; Scetbon et al., 2025), ensuring that SRON's memory footprint remains comparable to these baselines.

**Main Results.** We evaluate SRON and SRON$^\dagger$ on pre-training LLaMA models ranging from 60M to 1.3B parameters, focusing on final validation perplexity (PPL) and estimated optimizer memory usage (BF16). The results are presented in Figure 1 and Table 2.

Our experiments demonstrate that **SRON achieves superior performance while consuming less memory and enabling faster training**. Across all tested models, SRON consistently attains lower or comparable final validation PPL compared to baseline optimizers, while maintaining significantly reduced memory usage. Specifically, for LLaMA-1.3B, SRON reduces total training memory by 60% and optimizer state memory by approximately 90% relative to Adam, while shortening total training time by 67%. Furthermore, SRON decreases optimizer memory consumption by 75% compared to other memory-efficient optimizers, including GaLore and APOLLO.

Among optimizers with zero auxiliary states (SGD, SignSGD, and SRON$^\dagger$), only SRON$^\dagger$ achieves performance comparable to Adam. Compared with SRON, SRON$^\dagger$ completely eliminates optimizer state memory overhead, though at the cost of a modest performance drop. This drop arises because embedding layer parameters are one-dimensional; in this setting, row normalization degenerates into global normalization, thereby discarding dimensional information. Overall, this highlights a memory–performance trade-off, with SRON$^\dagger$ representing an attractive option for training under strict resource constraints, and successfully achieving a fully global state-free optimizer design.

**Scaling to LLaMA-3B/7B.** For larger models, we compare SRON with 8-bit Adam (Dettmers et al., 2021), GaLore, APOLLO, and APOLLO-Mini, employing gradient checkpointing (Chen

Table 3: **Pre-training LLaMA-3B and LLaMA-7B models on the C4 dataset.** We report the validation PPL across training steps, wall-clock training time, and estimated memory usage for optimizer states. We train LLaMA-3B on 16 NVIDIA RTX 3090 GPUs (24GB each), and LLaMA-7B on 4 NVIDIA H100 GPUs (80GB each).

| Models | Methods | 30K | 60K | 90K | 120K | 150K | Mem. | Time | Tokens/s |
|--------|---------|-----|-----|-----|------|------|------|------|----------|
| LLaMA-3B | 8-bit Adam | 19.11 | 15.88 | 14.67 | 14.31 | - | 5.30G | 297.3h | 14.9K |
| | GaLore | 18.44 | 15.98 | 14.90 | 14.73 | - | 3.91G | 143.7h | 37.4K |
| | APOLLO | 19.49 | 15.40 | 14.09 | 13.75 | - | 3.91G | 125.7h | 37.8K |
| | APOLLO-Mini | 18.87 | 15.84 | 14.45 | 14.10 | - | 0.66G | 116.5h | 38.6K |
| | **SRON** | 16.06 | 13.58 | 12.28 | **11.92** | - | 0.65G | 114.4h | 39.4K |
| LLaMA-7B | 8-bit Adam | 18.79 | 15.71 | 14.14 | 13.38 | 13.23 | 13.5G | 285.5h | 19.9K |
| | GaLore | 18.35 | 15.63 | 14.33 | 13.77 | 13.69 | 9.40G | 341.8h | 21.0K |
| | APOLLO | 18.10 | 14.91 | 13.34 | 12.77 | 12.59 | 9.40G | 270.1h | 21.1K |
| | APOLLO-Mini | 18.59 | 15.27 | 13.64 | 12.90 | 12.73 | 1.05G | 268.0h | 21.2K |
| | **SRON** | 15.76 | 13.32 | 12.02 | 11.23 | **11.03** | 1.04G | 265.5h | 21.4K |
| Tokens seen | | 3.9B | 7.8B | 11.7B | 15.7B | 19.6B | | | |

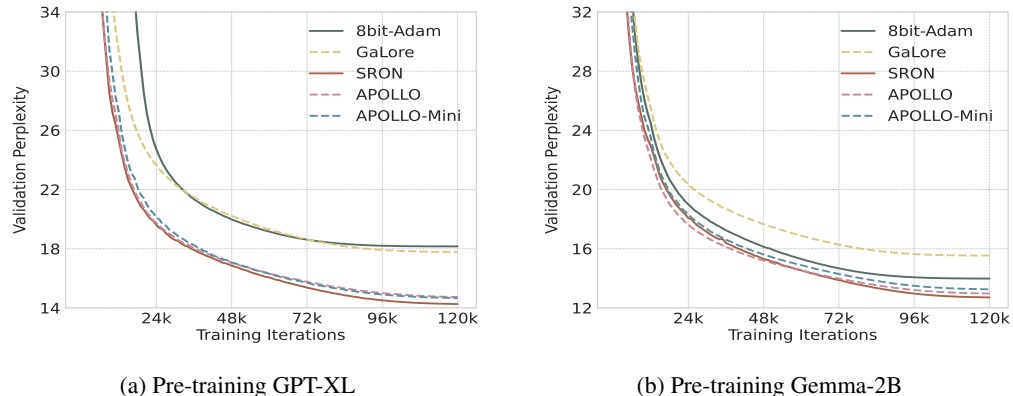

(a) Pre-training GPT-XL

(b) Pre-training Gemma-2B

Figure 3: Evaluating SRON in GPT-XL and Gemma-2B.

et al., 2016) to reduce memory usage. Due to limited computational resources, MUON is excluded from these experiments. The results, presented in Table 3, demonstrate that SRON achieves lower validation PPL under the same token budget while maintaining a smaller memory footprint. On LLaMA-3B with 16 NVIDIA 3090 GPUs, SRON reduces total training time by approximately 67% compared to 8-bit Adam and by 9% compared to SVD-free methods such as APOLLO, while also achieving lower PPL. Notably, in this experimental setting, SRON allows a batch size of 32, while 8-bit-Adam is limited to 8. On LLaMA-7B using NVIDIA H100 GPUs, SRON provides 7% faster training and reduces optimizer memory usage by 92%.

## 6.2 Additional Investigations and Ablation Studies

This section presents additional experimental results to further validate the effectiveness of SRON. We extend our evaluation beyond LLaMA to other architectures and conduct ablation studies on alternative normalization methods. We also examine SRON's performance under challenging settings, including long sequence lengths, extended training durations, small-batch training, and sensitivity to the hyperparameters $lr$ and $\alpha$. In addition, we assess SRON in downstream fine-tuning tasks. Comprehensive results are provided in Appendix B.

**Generalization to GPT and Gemma Models.** We extend SRON to additional architectures on the C4 dataset, including GPT-XL (1.5B) (Radford et al., 2019) and Gemma-2B (Team et al., 2024). Experimental results are shown in Figure 3. During the early stages of GPT-XL training, SRON and APOLLO exhibit similar validation PPL. However, as training progresses, SRON gradually outperforms APOLLO. A similar trend is observed with Gemma-2B: SRON surpasses APOLLO in the later stages once the number of training tokens increases. These findings further demonstrate

SRON's effectiveness across architectures beyond LLaMA, establishing it as a highly competitive optimizer for LLM pre-training.

**Ablation Study on Normalization.** We perform ablation experiments to compare different normalization methods. Specifically, we define **column-wise normalization** as

$$\mathbf{V}_{j,j}^t = \left(\sqrt{\frac{1}{m}\sum_{i=1}^m (\mathbf{G}_{i,j}^t)^2 + \epsilon}\right)^{-1}, \quad \tilde{\mathbf{G}}^t = \mathbf{G}^t \mathbf{V}^t,$$

and **tensor-wise (RMS) normalization** as

$$v^t = \left(\sqrt{\frac{1}{mn}\sum_{i=1}^m \sum_{j=1}^n (\mathbf{G}_{i,j}^t)^2 + \epsilon}\right)^{-1}, \quad \tilde{\mathbf{G}}^t = v^t \cdot \mathbf{G}^t.$$

These methods are compared to the **row-wise normalization** introduced in Algorithm 1. Additionally, we include **Lion** (Chen et al., 2023) as a baseline normalization method.

For our experiments, we pre-train LLaMA-60M and LLaMA-130M models on the C4 dataset. Results are shown in Figure 4. As illustrated, row-wise normalization in Algorithm 1 consistently outperforms both column-wise and tensor-wise (RMS) normalization, as well as the baseline methods. This outcome supports the observation findings in Section 4, which indicate that significant scale discrepancies exist across gradient rows during LLM training. Row-wise normalization, as defined in Eq. equation 2, effectively mitigates these discrepancies by aligning the scales of all rows, thereby enhancing optimization performance.

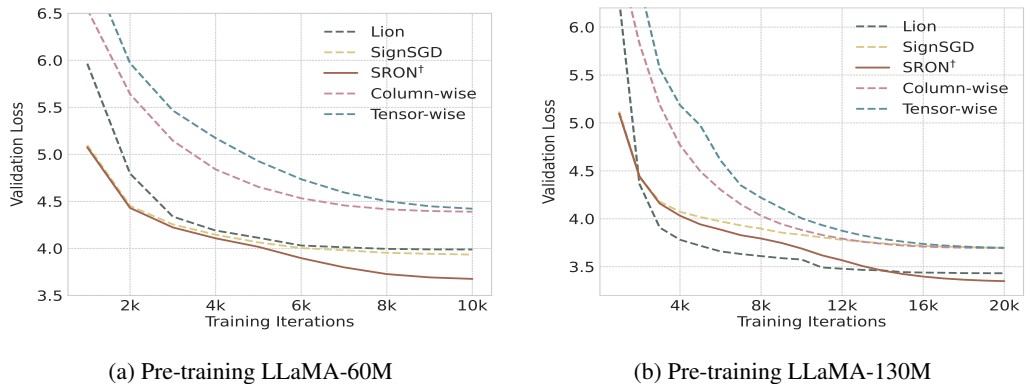

(a) Pre-training LLaMA-60M          (b) Pre-training LLaMA-130M

Figure 4: Ablation study on the normalization methods.

## 7 CONCLUSION

In this paper, we introduce **SGD with Row-wise Normalization (SRON)**, a state-free variant of SGD designed for memory-efficient LLM training. Unlike conventional normalization methods that flatten the entire multidimensional gradient into a single vector, SRON performs structure-aware normalization at the row level. This approach explicitly addresses gradient scale disparities within the Attention module, effectively eliminating inconsistencies in row-wise gradient magnitudes during training. Consequently, SRON enables fully state-free parameter updates while reducing memory overhead by up to 90%–100%. Theoretically, we establish convergence guarantees under non-convex $L$-Lipschitz smoothness conditions, ensuring the soundness of SRON for modern LLM architectures. Empirically, SRON demonstrates clear advantages over existing optimizers, including Adam and recent memory-efficient methods, in terms of training speed, memory usage, and final performance. Evaluations across diverse architectures—such as LLaMA, GPT, and Gemma—show that SRON consistently matches or outperforms strong baselines, highlighting its effectiveness for LLM training.

## ETHICS STATEMENT

This paper presents work aimed at advancing the field of Machine Learning. No human subjects, sensitive personal data, or potentially harmful applications are involved. There are no ethical issues associated with this work.

## REPRODUCIBILITY STATEMENT

The complete proofs of all theorems are provided in the appendix. All code and scripts required to reproduce the experiments are included in the supplementary materials. The models and datasets used in our study are publicly available. Detailed experimental settings, hyperparameters, and evaluation protocols are described in the main paper and appendix. The source code for this work will be made publicly available upon publication.

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

## A  LEMMAS AND PROOFS

**Lemma A.1.** *For any $t \geq 1$, let $\mathbf{W}^t$ be the parameters obtained by SRON in Algorithm 1 after $t$-th iteration. Then*

$$\|\mathbf{W}^{t+1} - \mathbf{W}^t\|^2 \leq \eta_t^2 mn.$$

*Proof.* From the update rule in Eq. equation 3, we have

$$\|\mathbf{W}^{t+1} - \mathbf{W}^t\|^2 = \eta_t^2 \|\mathbf{V}^t \mathbf{G}^t\|^2 = \eta_t^2 \sum_{i=1}^m (\mathbf{V}_{i,i}^t)^2 \sum_{j=1}^n (\mathbf{G}_{i,j}^t)^2.$$

Leverage the definition of $\mathbf{V}$ in Eq. equation 2, we obtain

$$\eta_t^2 \sum_{i=1}^m (\mathbf{V}_{i,i}^t)^2 \sum_{j=1}^n (\mathbf{G}_{i,j}^t)^2 \leq \eta_t^2 \sum_{i=1}^m \left( \frac{1}{n} \sum_{j=1}^n (\mathbf{G}_{i,j}^t)^2 \right)^{-1} \left( \sum_{j=1}^n (\mathbf{G}_{i,j}^t)^2 \right) = \eta_t^2 mn.$$

Then the proof is completed. $\square$

### A.1  PROOF OF THEOREM 5.4

*Proof.* Leveraging the $L$-Lipschitz smooth property, we have

$$f(\mathbf{W}^{t+1}) \leq f(\mathbf{W}^t) + \langle \nabla f(\mathbf{W}^t), \mathbf{W}^{t+1} - \mathbf{W}^t \rangle + \frac{L}{2} \|\mathbf{W}^{t+1} - \mathbf{W}^t\|^2.$$

From the update rule $\mathbf{W}^{t+1} = \mathbf{W}^t - \eta_t \mathbf{V}^t \mathbf{G}^t$, we get

$$f(\mathbf{W}^{t+1}) \leq f(\mathbf{W}^t) - \eta_t \langle \nabla f(\mathbf{W}^t), \mathbf{V}^t \mathbf{G}^t \rangle + \frac{L}{2} \|\mathbf{W}^{t+1} - \mathbf{W}^t\|^2.$$

Taking conditional expectation $\mathbb{E}[\cdot \mid \mathcal{F}^t]$ on both sides:

$$\mathbb{E}[f(\mathbf{W}^{t+1}) \mid \mathcal{F}^t] \leq f(\mathbf{W}^t) - \eta_t \mathbb{E}[\langle \nabla f(\mathbf{W}^t), \mathbf{V}^t \mathbf{G}^t \rangle \mid \mathcal{F}^t] + \frac{L}{2} \mathbb{E}[\|\mathbf{W}^{t+1} - \mathbf{W}^t\|^2 \mid \mathcal{F}^t].$$

By Lemma A.1, we have

$$\mathbb{E}[\|\mathbf{W}^{t+1} - \mathbf{W}^t\|^2 \mid \mathcal{F}^t] \leq \eta_t^2 mn.$$

Consider the inner product term. Using the unbiasedness of the stochastic gradient $\mathbb{E}[\mathbf{g}_i^t \mid \mathcal{F}^t] = [\nabla f(\mathbf{W}^t)]_i$, we have

$$\mathbb{E}[\langle \nabla f(\mathbf{W}^t), \mathbf{V}^t \mathbf{G}^t \rangle \mid \mathcal{F}^t] = \sum_{i=1}^m \mathbb{E}[\mathbf{V}_{i,i}^t \langle [\nabla f(\mathbf{W}^t)]_i, \mathbf{g}_i^t \rangle \mid \mathcal{F}^t]$$

$$= \sum_{i=1}^m \mathbb{E}[\mathbf{V}_{i,i}^t \| [\nabla f(\mathbf{W}^t)]_i \|^2 \mid \mathcal{F}^t] \geq c_1 \|\nabla f(\mathbf{W}^t)\|^2,$$

where $c_1 := \min_i \mathbb{E}[\mathbf{V}_{i,i}^t \mid \mathcal{F}^t] > 0$.

Incorporating the above inequalities, we get

$$\mathbb{E}[f(\mathbf{W}^{t+1}) \mid \mathcal{F}^t] \leq f(\mathbf{W}^t) - \eta_t c_1 \|\nabla f(\mathbf{W}^t)\|^2 + \frac{L\eta_t^2 mn}{2}.$$

Taking the global expectation

$$\mathbb{E}[f(\mathbf{W}^{t+1})] \leq \mathbb{E}[f(\mathbf{W}^t)] - \eta_t c_1 \mathbb{E}\|\nabla f(\mathbf{W}^t)\|^2 + \frac{L\eta_t^2 mn}{2}.$$

Summing from $t = 1$ to $T$, we obtain

$$c_1 \sum_{t=1}^T \eta_t \mathbb{E}\|\nabla f(\mathbf{W}^t)\|^2 \leq \mathbb{E}[f(\mathbf{W}^1)] - f^* + \frac{Lmn}{2} \sum_{t=1}^T \eta_t^2.$$

Choosing $\eta_t = \frac{\eta_0}{\sqrt{t}}$, we have

$$\sum_{t=1}^{T} \eta_t \geq 2\eta_0(\sqrt{T+1} - 1), \quad \sum_{t=1}^{T} \eta_t^2 \leq \eta_0^2(1 + \ln T).$$

Therefore,

$$\min_{1 \leq t \leq T} \mathbb{E}\|\nabla f(\mathbf{W}^t)\|^2 \leq \frac{\mathbb{E}[f(\mathbf{W}^1)] - f^*}{2c_1\eta_0(\sqrt{T+1} - 1)} + \frac{Lmn\eta_0(1 + \ln T)}{4c_1(\sqrt{T+1} - 1)}.$$

In other words,

$$\min_{1 \leq t \leq T} \mathbb{E}\|\nabla f(\mathbf{W}^t)\|^2 = \mathcal{O}\left(\frac{\ln T}{\sqrt{T}}\right).$$

The proof is completed. □

## B SUPPLEMENTARY EXPERIMENTS

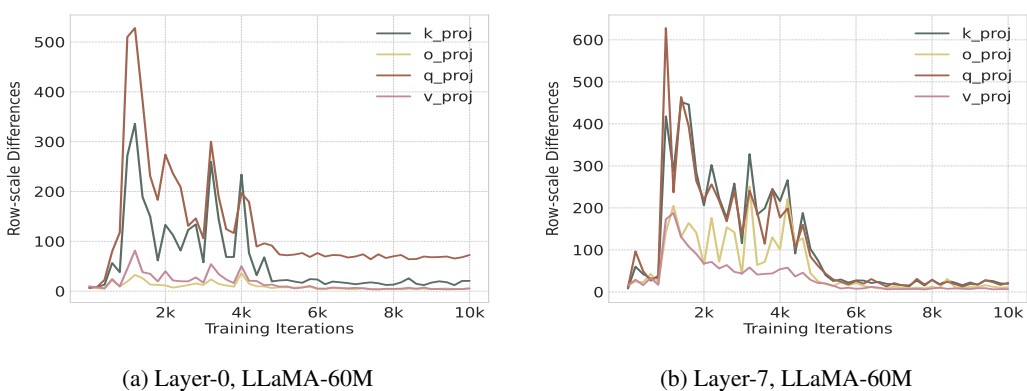

(a) Layer-0, LLaMA-60M

(b) Layer-7, LLaMA-60M

Figure 5: Row-norm scale differences of gradients in attention modules $\mathbf{W}_Q, \mathbf{W}_K, \mathbf{W}_V, \mathbf{W}_O$ on LLaMA-60M.

### B.1 LONG-CONTEXT TRAINING

To evaluate SRON under extended context lengths, we train LLaMA models (60M–350M) with sequences of length 512 and 1024 while keeping total tokens per batch fixed at 131K. Table 4 shows that SRON maintains superior PPL compared to baselines. We can see that when the sequence length increases, all methods, including SRON, show a certain degree of performance degradation, which is especially pronounced in GaLore and APOLLO-Mini. The reason is that our experimental configuration actually decreased the iteration-independent batch size. Nevertheless, SRON continues to outperform the tested baselines, showcasing its strength in long-sequence training.

### B.2 LONG-TERM TRAINING

We assess SRON with extended training schedules on LLaMA-60M and LLaMA-130M, using 180B and 390B tokens, respectively ( 30 tokens per parameter, exceeding the Chinchilla scaling recommendation (Hoffmann et al., 2022)). We present the learning curve in Figure 6. Across these prolonged training regimes, SRON consistently preserves lower PPL, demonstrating its ability to handle massive datasets without performance degradation.

### B.3 SMALL-BATCH TRAINING

An immediate observation is that SRON, being entirely stateless, cannot leverage moving averages to reduce gradient noise as Adam and other momentum-based optimizers do (Cutkosky & Mehta,

Table 4: **Evaluation of SRON in long-sequence training configurations.** We evaluate SRON with a sequence length of 512 and 1024. Final validation PPLs are reported. SRON consistently demonstrates strong performance, maintaining its effectiveness across extended sequence lengths.

| Methods | 60M | | 130M | | 350M | |
|---|---|---|---|---|---|---|
| | 512 | 1024 | 512 | 1024 | 512 | 1024 |
| Adam | 34.52 | 37.52 | 25.95 | 28.68 | 19.95 | 22.02 |
| GaLore | 35.25 | 38.09 | 27.19 | 29.51 | 19.92 | 21.73 |
| APOLLO | 32.02 | 34.04 | 24.04 | 25.93 | 17.26 | 18.77 |
| APOLLO-Mini | 32.76 | 35.15 | 24.58 | 26.51 | 17.75 | 19.39 |
| **SRON** | **30.79** | **33.37** | **23.11** | **25.07** | **16.67** | **18.33** |
| Training Tokens | 1.3B | | 2.6B | | 7.8B | |

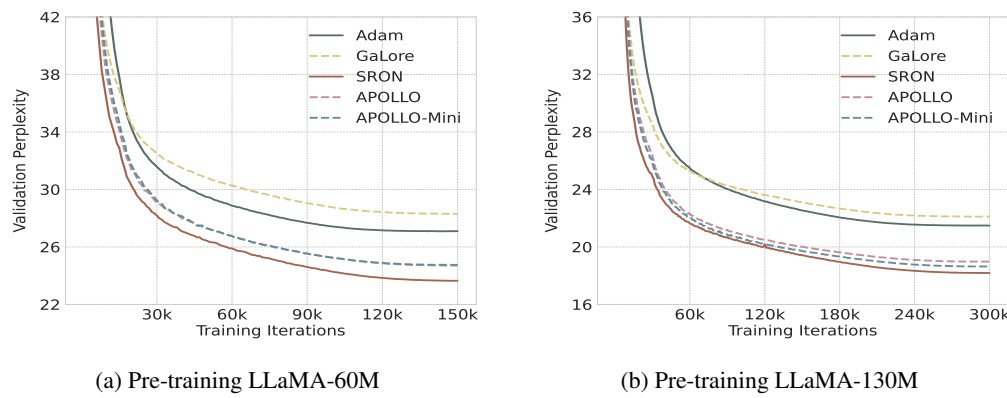

(a) Pre-training LLaMA-60M    (b) Pre-training LLaMA-130M

Figure 6: Long-term training on LLaMA-60M and 130M.

2020; Crawshaw et al., 2022), particularly in small-batch regimes where gradient noise is more pronounced. To evaluate SRON under such conditions, we conduct small-batch training with total batch sizes of 128 (one-eighth or one-fourth of the original setting, respectively) and increase the number of training iterations eightfold or fourfold to keep the total number of training tokens unchanged. All other hyperparameters remain consistent with the main experiments.

The results, summarized in Table 5, show that SRON continues to achieve performance comparable to or exceeding that of other baselines. Relative to the experimental setup in the main text in Table 2, the model PPL achieved by each tested method shows some decline. Under extremely small batch sizes, such as 64, SRON shows a performance degradation, performing worse than APOLLO and APOLLO-Mini, but still better than Full-Adam and GaLore. This suggests that the absence of optimizer states in SRON leads to inaccurate estimation of row-normalization statistics under high-noise conditions. In scenarios with a high proportion of noise, a sliding window can be considered to mitigate the disturbance caused by the gradients of the current batch, which we leave for future work.

### B.4 MEMORY-EFFICIENT FINE-TUNING

Compared to pre-training, fine-tuning is generally more practical for engineers and researchers, as it requires substantially fewer computational resources. In this section, we evaluate SRON in fine-tuning experiments using the RoBERTa-Large model (Liu, 2019) on the GLUE benchmark (Wang et al., 2018). We compare SRON against several baselines, including **Adam** (Kingma & Ba, 2014), **APOLLO** (Zhu et al., 2024), and **GaLore** (Zhao et al., 2024). In addition, we include **LoRA** (Hu et al., 2022), a widely adopted memory-efficient fine-tuning method. We adopt a sequence length of 256 for all tasks and set the epoch to 3. To ensure fairness, we tune the learning rate for each method, selecting the best value from $\{1.0e-5,\ 2.5e-5,\ 5.0e-5,\ 1.0e-4,\ 1.5e-4,\ 2.0e-4\}$. For all low-rank methods, we set $r = 4$.

Table 5: **Evaluation of SRON with a batch size of 64 and 128.** We report the final validation PPL.

| Methods | 60M | | 130M | |
|---|---|---|---|---|
| | 64 | 128 | 64 | 128 |
| Adam | 37.31 | 35.48 | 25.95 | 27.06 |
| GaLore | 38.38 | 36.34 | 27.19 | 28.03 |
| APOLLO | **32.34** | 31.37 | **24.31** | 24.77 |
| APOLLO-Mini | 32.84 | 31.92 | 24.58 | 25.69 |
| **SRON** | 33.64 | **31.29** | 25.28 | **23.99** |
| Training Tokens | 1.3B | | 2.6B | |

As shown in Table 6, SRON achieves performance that is comparable to, or surpasses, the baselines across a range of downstream LLM tasks. These results demonstrate SRON's effectiveness beyond pre-training and suggest that it can serve as a unified, memory-efficient optimization method applicable throughout the LLM training pipeline.

Table 6: **Evaluating SRON for memory-efficient fine-tuning on the GLUE benchmark (higher is better)**, using a pre-trained RoBERTa-Large model. We report overall (matched and mismatched) accuracy for MNLI, Matthew's correlation coefficient for CoLA, Pearson correlation for STS-B, and classification accuracy for all other tasks.

| Methods | CoLA | STS-B | MRPC | RTE | SST2 | MNLI | QNLI | QQP | Avg. |
|---|---|---|---|---|---|---|---|---|---|
| Full-Adam | 64.85 | 91.60 | 92.79 | 78.81 | 96.44 | 90.51 | 94.43 | 91.90 | 87.66 |
| LoRA | **64.32** | 90.68 | 91.39 | 77.72 | 95.98 | **90.57** | **94.78** | **90.93** | 87.04 |
| GaLore | 62.52 | 91.18 | 90.94 | 77.11 | **96.10** | 90.27 | 94.21 | 89.99 | 86.54 |
| APOLLO | 61.13 | 91.66 | **92.14** | 80.14 | 95.06 | 89.85 | 93.61 | 89.27 | 86.58 |
| **SRON** | 63.08 | **91.91** | 91.76 | **83.03** | 94.72 | 89.74 | 93.78 | 89.59 | **87.20** |

# C    IMPLEMENTATION DETAILS

## C.1    NETWORK SETUP

In this section, we describe the model architectures used in our experiments, including Large Language Model Meta AI (LLaMA) (Touvron et al., 2023b), Generative Pre-trained Transformer (GPT) (Radford et al., 2019), and Gemma (Team et al., 2024). To ensure fairness, we adopt the LLaMA architectural details reported in prior work (Zhao et al., 2024), which follows the design choices of RMSNorm and SwiGLU activations (Touvron et al., 2023b;a). Table 7 summarizes the architectural hyperparameters of LLaMA models of different sizes, as well as those of GPT and Gemma.

## C.2    HYPERPARAMETERS

For the $lr$ hyperparameter of the tested baselines, we tune the learning rates of Adam, MUON, Lion, SGD, and SignSGD, selecting the best value from $\{1.0e-5, 5.0e-5, 1.0e-4, 5.0e-4, 1.0e-3, 5.0e-3, 1.0e-2, 5.0e-2, 1.0e-1\}$. For memory-efficient baselines (GaLore, APOLLO, APOLLO-Mini), we follow the learning rates and scaling coefficient $\alpha$ suggested in the respective original works.

For our method (SRON), we apply row-wise normalization to all linear projection weights, while the remaining parameters are optimized using Adam. Following prior work (Zhao et al., 2024; Chen et al., 2024; Zhu et al., 2024; Wen et al., 2025; Ma et al., 2024; Liu et al., 2025), we adopt a global learning rate to control all modules and introduce a scaling factor $\alpha$ for the SRON-updated parameters. Specifically, the effective learning rate for Adam-updated parameters is $lr$, whereas for

Table 7: Architecture hyperparameters of different models for pre-training. Batch size and training data amount are specified in tokens.

| Models | Params | Hidden | Intermediate | Heads | Layers | Iteration | Tokens |
|--------|--------|--------|--------------|-------|--------|-----------|--------|
| LLaMA | 60M | 512 | 1376 | 8 | 8 | 10K | 1.3B |
|  | 130M | 768 | 2048 | 12 | 12 | 20K | 2.6B |
|  | 350M | 1024 | 2736 | 16 | 24 | 60K | 7.8B |
|  | 1B | 2048 | 5461 | 24 | 32 | 100K | 13.1B |
|  | 3B | 2560 | 6848 | 32 | 32 | 120K | 15.7B |
|  | 7B | 4096 | 11008 | 32 | 32 | 150K | 19.7B |
| GPT | 1.5B | 1600 | - | 25 | 48 | 120K | 15.7B |
| Gemma | 2B | 2048 | 16384 | 8 | 18 | 120K | 15.7B |

SRON-updated parameters it is $lr \times \alpha$. In line with SWAN (Ma et al., 2024), we employ a *lazy-tuning* strategy, setting hyperparameters without extensive search to reduce the risk of unfair comparisons arising from over-tuning. Specifically, we test $lr \in \{1.0e-2, \ 2.0e-2\}, \alpha \in \{5.0e-2, \ 1.0e-1\}$. This tuning approach guarantees that the SRON module's effective learning rate stays at $0.001$, identical to Adam's default learning rate. We present the corresponding experimental results in Table 9. For the SRON$^{\dagger}$ variant, where SRON is applied to all model parameters, the scaling factor $\alpha$ is no longer required. Instead, we perform a grid search over $\{1.0e-4, \ 5.0e-4, \ 1.0e-3, \ 5.0e-3\}$ to determine the optimal learning rate. We present all the hyperparameters in Table 8.

Table 8: Hyperparameters $(lr, \alpha)$ for pre-training LLaMA models. SRON applies the identical hyperparameters for all models, while other methods need to lower the learning rate during LLaMA-3B training to avoid loss explosion in our test.

| Models | 60M | | 130M | | 350M | | 1B | | 3B | | 7B | |
|--------|-----|-----|------|-----|------|-----|-----|-----|-----|-----|-----|-----|
| Hyperparameters | $lr$ | $\alpha$ | $lr$ | $\alpha$ | $lr$ | $\alpha$ | $lr$ | $\alpha$ | $lr$ | $\alpha$ | $lr$ | $\alpha$ |
| Adam (8bit) | 5.0e-3 | - | 1.0e-3 | - | 1.0e-3 | - | 5.0e-4 | - | 5.0e-4 | - | 5.0e-4 | - |
| MUON | 5.0e-3 | - | 5.0e-3 | - | 1.0e-3 | - | 1.0e-3 | - | - | - | - | - |
| Lion | 1.0e-4 | - | 1.0e-4 | - | 1.0e-4 | - | 1.0e-4 | - | - | - | - | - |
| GaLore | 1.0e-2 | 0.25 | 1.0e-2 | 0.25 | 1.0e-2 | 0.25 | 1.0e-2 | 0.25 | 5.0e-3 | 0.25 | 1.0e-2 | 0.25 |
| APOLLO | 1.0e-2 | 1.0 | 1.0e-2 | 1.0 | 1.0e-2 | 1.0 | 1.0e-2 | 1.0 | 5.0e-3 | 1.0 | 1.0e-2 | 128 |
| APOLLO-Mini | 1.0e-2 | 128 | 1.0e-2 | 192 | 1.0e-2 | 128 | 1.0e-2 | 128 | 5.0e-3 | 128 | 1.0e-2 | 128 |
| **SRON** | 2.0e-2 | 0.05 | 2.0e-2 | 0.05 | 2.0e-2 | 0.05 | 2.0e-2 | 0.05 | 2.0e-2 | 0.05 | 2.0e-2 | 0.05 |
| SGD | 1.0e-1 | - | 1.0e-1 | - | 1.0e-1 | - | 1.0e-1 | - | - | - | - | - |
| SignSGD | 1.0e-3 | - | 1.0e-3 | - | 1.0e-3 | - | 1.0e-3 | - | - | - | - | - |
| **SRON$^{\dagger}$** | 1.0e-3 | - | 1.0e-3 | - | 1.0e-3 | - | 5.0e-4 | - | - | - | - | - |
| $r$ for low-rank methods | 128 | | 192 | | 256 | | 512 | | 640 | | 1024 | |

All experiments are conducted with a global batch size of 512 and a sequence length of 256, yielding $512 \times 256 = 131K$ tokens per iteration. We adopt BF16 precision for model parameters, gradients, and optimizer states to reduce memory consumption. The learning rate schedule follows (Zhao et al., 2024): a linear warmup over the first 10% of training iterations, followed by cosine decay to 10% of the base learning rate. For reproducibility, all training runs are performed with the random seed fixed at 42.

## C.3 MEMORY ESTIMATION

For memory estimation, we compute optimizer memory layer-by-layer using the proposed memory consumption reported in Table 1. Specifically, we isolate memory overhead attributable to model parameters and optimizer states, while excluding factors such as batch size and PyTorch's memory caching and fragmentation behavior (Paszke et al., 2017). The code used for memory estimation is provided in the supplementary materials.

Table 9: Evaluation of $lr$ and $\alpha$ hyperparameter combinations for SRON on LLaMA-60M/130M. We report the final validation PPL.

| $lr$ | $\alpha$ | LLaMA-60M | LLaMA-130M |
|---|---|---|---|
| 0.01 | 0.1 | 30.37 | 22.64 |
|  | 0.05 | 30.83 | 22.88 |
| 0.02 | 0.1 | 30.08 | 22.52 |
|  | 0.05 | **29.91** | **22.51** |

As a representative example, the LLaMA-1.3B model contains approximately 1,339.08M parameters, of which 1,207.91M are updated with SRON and 191.17M with Adam. This corresponds to $131.17M \times 2\text{Bytes} \times 2 = 524.68\text{MB} = 0.52\text{GB}$ for storing optimizer states in the Adam portion. Since SRON is state-free, the total optimizer memory consumption for training LLaMA-1.3B with SRON is only 0.52GB, while SRON$^{\dagger}$ requires effectively 0.00GB.

### C.4  GPU Communication

Gradient communication across GPUs remains a critical bottleneck in distributed training. Compared with gradient orthogonalization or whitening methods such as SWAN (Ma et al., 2024), MUON (Liu et al., 2025), and SinkGD (Scetbon et al., 2025), SRON's gradient preprocessing does not require storing the entire gradient matrix and is compatible with gradient sharding, which in principle can integrate with existing sharded communication methods. As GPU communication overhead is not the primary focus of this work, we do not discuss it in detail here.

### C.5  Future Works

Several directions remain open for further exploration:

1. The comparison of SRON and SRON$^{\dagger}$ performance, along with the hybrid optimizer strategies in other memory-efficient approaches, indicates that Adam is still irreplaceable for training nonlinear layers. Future work includes exploring methods to compress the memory usage required for training these parameters.

2. Under high-noise conditions, SRON's row normalization can be significantly affected; in such cases, incorporating a sliding window over batches may help mitigate the noise impact.

3. Extending SRON to other model families, including ViTs (Dosovitskiy et al., 2021) and Diffusion Models (Song et al., 2021). Since these models often contain numerous three-dimensional or even four-dimensional parameter modules, applying SRON to these parameters requires additional design considerations, which we leave for future work.

4. Investigating SRON's compatibility with advanced parallel training strategies beyond torch.DDP, such as model parallelism (Shazeer et al., 2018) and pipeline parallelism (Huang et al., 2019).

5. Due to computational resource limitations, the largest model we used to evaluate SRON's performance was 7B, which is sufficient from an academic perspective. However, its performance on ultra-large models and trillion-token scales still awaits industrial validation, as training such models often requires over 1000 GPUs and is more prone to adverse training environments.

## D  Use of Large Language Models

Large language models (LLMs) were used in preparing this manuscript solely for linguistic assistance, including polishing wording, improving clarity, and refining phrasing. LLMs were not

involved in generating scientific ideas, designing experiments, analyzing data, or drawing conclusions. All technical contributions, experimental work, and findings are entirely the responsibility of the authors.

