# OpenReview forum: "SRON: State-free LLM Training via Row-wise Gradient Normalization"
_ICLR.cc/2026/Conference — ICLR 2026 Conference Withdrawn Submission_

### Official Review · Reviewer_fFgb · 2025-10-19

**Soundness:** 2
**Presentation:** 3
**Contribution:** 1
**Rating:** 2
**Confidence:** 4

**Summary:**

The paper proposes SRON (SGD with Row-wise Normalization), a stateless optimizer that rescales each row of weight-matrix gradients by its RMS norm (Eq. (2)) and then performs an SGD step (Alg. 1). The motivation is that attention projections exhibit large row-to-row scale disparities; normalizing per row should stabilize updates. The paper proves an $\mathcal{O}(\log T / \sqrt{T})$ non-convex convergence rate and reports pre-training results on LLaMA/GPT/Gemma from 60M to 7B params. Empirically, SRON is claimed to reduce optimizer memory to ~0 and to match/outperform Adam and recent memory-efficient baselines, with large wall-clock speedups on some setups.

**Strengths:**

1. Simple, practical idea: Row-wise RMS scaling is easy to implement, state-free, and inexpensive
2. Clear algorithm & rate: Algorithm 1 is explicit, and the non-convex rate matches standard adaptive methods.
3. Experiment results: experiments shows the matching of the proposed approach with standard optimizers such as Adam

**Weaknesses:**

(please reply to the questions section directly, where I include the details of the weaknesses)
1. There are closely related and very recent works the paper does not cite or discuss.
2. The motivation is a bit vague
3. Experimental fairness / clarity

**Questions:**

1. Missing important related works [1], [2] and [3]: Adafactor [1] already consider only keeping the row/column norm information to save the memory of Adam, also it reduces the memory greatly; Scion [2] also proposes to consider the row/column normalization for certain layers (but they still have momentums); Most importantly, Scale [3] already consider row/column wise normalization and remove the momentums. Why didn't the author discuss the relation of the work to these related works? The methodology contribution seems not new here.
2. Is SRON normalizing w.r.t. the input or output dim? The $m$ and $n$ are quite confusing in the paper since it's not specified whether $m$ is the input or output dimension. In particular, it is observed in [3] that column-wise normalization is better than row-wise normalization, where they assume $W\in\mathbb{R}^{m\times n}$ and $m$ is the input dimension and $n$ is the output dimension. If their $m$ is the same as the $n$ in this work, then it means that the result in this paper is no different from [3]; on the other hand if their $m$ is the same as the $m$ in this work, then it means that this work is having opposite conclusion to [3]. This is very important and needs detailed clarification.
3. Continue above discussion, the paper's motivation is not that clear. In Figure 2 it is shown that row-wise differences makes it necessary to have row wise normalization. But this is not sufficient at all. How about the column-wise differences? How about the singular value differences? Why won't this motivate other form of normalization? Also after adding row-wise normalization, it is not clarified if the phenomenon in Fig 2 is mitigated.
4. Where is “SRON$^\dagger$” defined? Is it defined on line 353? But line 353 is using adam some modules, how come “SRON$^\dagger$” with 0.0G memory in Table 2? Or maybe the opposite, that “SRON$^\dagger$” is the one described in Algorithm 1?
5. Some typos: Line 161 “*employees* block coordinate descent” → employs; Line 190 "represnt" and "repesent"; Line 314, “we *access* SRON in models beyond LLaMA” → assess?; Multiple “Eq. equation X” (duplication): should fix to “Eq. (X)”.

References:
[1]. Shazeer, Noam, and Mitchell Stern. "Adafactor: Adaptive learning rates with sublinear memory cost." International Conference on Machine Learning. PMLR, 2018.
[2]. Pethick, Thomas, et al. "Training Deep Learning Models with Norm-Constrained LMOs." Forty-second International Conference on Machine Learning, 2025.
[3]. Glentis, Athanasios, et al. "A Minimalist Optimizer Design for LLM Pretraining." arXiv preprint arXiv:2506.16659 (2025).

---

### Official Review · Reviewer_dhZ3 · 2025-10-31

**Soundness:** 2
**Presentation:** 1
**Contribution:** 1
**Rating:** 2
**Confidence:** 5

**Summary:**

The paper focuses on reducing optimizer memory during LLM pretraining by using SGD with row-wise normalized gradients for the transformer layers and Adam for the first and last layer. It also experiments with applying row-wise normalized SGD to the entire network. It includes pretraining experiments on C4 across three model types, showing the effectiveness of the approach.

**Strengths:**

The paper includes a large number of experiments validating the effectiveness of row-wise normalized SGD.

The writing is clear and easy to follow.

Code is included in the submission, aiding reproducibility.

**Weaknesses:**

I find that the gradient normalization this paper explores to improve the SGD effectiveness (i.e. the main algorithm in this work, dubbed "SRON") has already been proposed in prior literature.

Specifically, [1] proposes SCALE, which uses "column-wise gradient normalization" along with SGD without momentum except for the last layer. I find that the normalization operation performed is the same as here, but [1] uses a different naming convention for the input and output of the weight matrices, thus naming the normalization column-wise instead of row-wise. In fact, I find that in Table 2, method "SRON†" here has almost identical results as in Table 2, method "column-wise" of [1]. Also, Table 5. of [1] shows that column-wise normalization (or equivalently row-wise based on opposite naming convention) combined with only last-layer momentum can be sufficient for LLM pretraining.

In addition, [2] explores a number of gradient normalization methods for SGD, including row- and column-wise. [2] also provides convergence analysis.

Therefore I cannot count row-wise gradient normalized SGD as a contribution of this paper. This limits novelty significantly, as this is the main algorithm explored. I find it confusing that the approach is renamed here as "SRON" and I would ask the authors to make clear what is originally proposed in this work and properly cite related literature (since [1] and [2] are not cited in the provided manuscript).

This work also proposes using Adam for the first and last layers for improved performance. However this introduces a tradeoff between memory usage and performance as well as further hyperparameters for tuning.

In addition, the explanation for row-wise's success is not entirely convincing ("discrepancies in gradient row-norm scales"). Although Figure 4. has an ablation comparing gradient normalization methods when used to all the layers (including the first and last layer), the explanation in Section 4 focuses on discrepancies in attention layers. It is therefore unclear how other normalization methods will perform if applied only to transformer layers and Adam is used for the first and last layers (as in "SRON"). If the performance difference between the different normalization types is much less significant, then this significantly weakens the explanation provided.

[1] "A Minimalist Optimizer Design for LLM Pretraining", 2025.
[2] "Training Deep Learning Models with Norm-Constrained LMOs", 2025.

**Questions:**

Have the authors tested other types of normalization (e.g. column-wise) for the transformer layers while using Adam for the first/last layer? Ablations on pretraining small models such as 60M and 130M LLaMA would be enough.

---

### Official Review · Reviewer_Cf8z · 2025-11-01

**Soundness:** 2
**Presentation:** 2
**Contribution:** 2
**Rating:** 6
**Confidence:** 3

**Summary:**

The paper proposes SRON, a state-free optimizer that applies row-wise gradient normalization for 2D parameter tensors (e.g., projection matrices) and updates them via SGD. Motivated by large row-norm disparities observed in attention modules, SRON scales rows by an RMS-style factor and claims strong memory savings (≈90–100% optimizer-state reduction) and up to 67% wall-clock speedups while matching or surpassing Adam and recent memory-efficient baselines on LLaMA/GPT/Gemma pre-training and GLUE fine-tuning. A convergence guarantee under non-convex L-smoothness is provided with rate $\(O(\ln T/\sqrt{T})\)$.

**Strengths:**

- Clarity and simplicity: structure-aware row-wise normalization (vs. tensor-wise flattening) is well motivated by measured row-norm variance in attention blocks; the algorithm is easy to implement.
- Strong empirical results with concrete systems outcomes: reduced optimizer memory, faster tokens/sec, and lower PPL across 60M–7B settings; ablations indicate row-wise > column-wise/tensor-wise normalization.
- Theory matches practice: a non-convex convergence rate $O(\ln T/\sqrt{T})$ complements the empirical story; the presentation of Algorithm 1 and comparisons to Adam/GaLore/APOLLO are generally clear.

**Weaknesses:**

- “State-free” positioning is diluted by the hybrid recipe (SRON on linear projections, Adam on others). Claims should be more precise about which parameters use states and how much of the model that covers at each scale.
- Fairness/tuning confounds: several baselines depend on sensitive hyperparameters (e.g., ranks, scaling factors). The paper uses “lazy tuning” for SRON but cites tuned settings for others; reported speedups may also partly stem from different feasible batch sizes.
- Edge cases and limitations are underexplored: SRON+ degrades on embeddings (1D degeneracy) and in very small-batch regimes; communication overheads of the row-wise statistics under DDP/sharding are not profiled.

**Questions:**

- Theorem 5.4: you introduce $c_1 := \min_i \mathbb{E}[V^t_{i,i}]>0$. Under what assumptions on per-row gradient moments does $c_1$ admit an explicit lower bound in terms of $\(n,\,\epsilon\)$ and second moments? How sensitive is the rate to the choice of $\epsilon$?
- Algorithm 1 scaling: SRON uses a global lr with a module scaling factor $\alpha$, while Adam parts use lr directly. Please report a 2D grid over $\{\text{lr}\}\times\{\alpha\}$ with tokens/sec and final PPL, and clarify whether $\alpha$ is layer-specific or shared; also confirm that Adam baselines were allowed equally favorable batch sizes/schedules.
- Hybrid “state-free” claim: for each model size, what fraction of parameters (and optimizer-state memory) actually run on Adam vs. SRON? Provide a per-layer breakdown and reproduce the 1.3B/3B/7B memory numbers with code to compute state bytes; can SRON† be stabilized on embeddings via a short sliding-window RMS (instead of single-batch) to avoid the 1D collapse?

---

### Official Review · Reviewer_grUh · 2025-11-01

**Soundness:** 2
**Presentation:** 3
**Contribution:** 2
**Rating:** 4
**Confidence:** 3

**Summary:**

This paper proposes SRON (SGD with Row-wise Normalization), a memory-efficient optimizer for training Large Language Models that eliminates optimizer state storage by applying row-wise normalization to gradients before parameter updates. The method is motivated by observations that gradients in attention modules exhibit extreme row-level scale disparities (up to 500-fold differences), which row-wise normalization addresses more effectively than global normalization approaches. SRON reduces optimizer memory by 90-100% compared to Adam and cuts training time by up to 67% on billion-parameter models, while consistently matching or outperforming Adam's validation perplexity.

**Strengths:**

1. This paper focus on an interesting and important problem, memory-efficient optimizer for LLM pre-training.

2. The proposed method is simple but efficient and the paper is also easy to follow.

3. This paper also provides extensive and thorough experimental validation, tests range from 60M to 7B parameters across multiple architectures (LLaMA, GPT, Gemma).

**Weaknesses:**

1. The motivation is clear, but the reason why we can use row-wise normalization in the proposed method and why the proposed method can achieve a better performance than current baseline methods is not very clear to me, how about column-wise normalization?

2. I think the gradient normalization in optimizer papers is not new, although the authors provide some updates. I think this will limit their contribution.

3. The paper lacks comparisons with several important recent baseline methods in the main results (such as Lion in table 2, muon and lion in table 3).

**Questions:**

1. I would like to ask whether the authors fully tune the hyper-parameters of the baseline methods.

2. I would like to ask whether the authors can provide more results on post-training. Because for most researchers, they can only afford post-training. In addition, I understand the authors provide the results on Glue. However, in my opinion, these tasks have been solved well, and maybe we should focus more on the challenging tasks, such as fine-tuning on mathematical reasoning, code generation and RL fine-tuning.

---

### Note · Authors · 2025-11-18

I have read and agree with the venue's withdrawal policy on behalf of myself and my co-authors.